# Genetically detoxified pertussis toxin displays near identical structure to its wild-type and exhibits robust immunogenicity

Salvador F. Ausar[1], Shaolong Zhu[2,3], Jessica Duprez[2], Michael Cohen[3,5], Thomas Bertrand[4], Valérie Steier[4], Derek J. Wilson [3], Stephen Li[5], Anthony Sheung[1], Roger H. Brookes [1], Artur Pedyczak[2], Alexey Rak[4] & D. Andrew James [2,3 ✉]

The mutant gdPT R9K/E129G is a genetically detoxified variant of the pertussis toxin (PTx) and represents an attractive candidate for the development of improved pertussis vaccines. The impact of the mutations on the overall protein structure and its immunogenicity has remained elusive. Here we present the crystal structure of gdPT and show that it is nearly identical to that of PTx. Hydrogen-deuterium exchange mass spectrometry revealed dynamic changes in the catalytic domain that directly impacted $NAD^+$ binding which was confirmed by biolayer interferometry. Distal changes in dynamics were also detected in S2-S5 subunit interactions resulting in tighter packing of B-oligomer corresponding to increased thermal stability. Finally, antigen stimulation of human whole blood, analyzed by a previously unreported mass cytometry assay, indicated broader immunogenicity of gdPT compared to pertussis toxoid. These findings establish a direct link between the conserved structure of gdPT and its ability to generate a robust immune response.

[1] Bioprocess Research and Development, Sanofi Pasteur Ltd., Toronto, ON M2R 3T4, Canada. [2] Analytical Sciences, Sanofi Pasteur Ltd., Toronto, ON M2R 3T4, Canada. [3] Center for Research in Mass Spectrometry, Department of Chemistry, York University, Toronto, ON M3J 1P3, Canada. [4] Research Platform, Sanofi R&D, Vitry-sur-Seine, 94400 Paris, France. [5] Present address: Fluidigm Corporation, Markham, ON L3R 4G5, Canada. ✉email: Andrew.James@sanofi.com

Pertussis, also known as whopping cough, is a highly contagious acute bacterial infection of the respiratory tract caused primarily by the bacterium *Bordetella pertussis*[1]. It is particularly challenging among vaccine-preventable diseases because neither infection nor vaccination confers life-long protection. The most effective strategy to reduce the number of cases of pertussis is vaccination and this has significantly decreased the burden of disease. However, despite decades of vaccination, the incidence of pertussis has begun to increase in many countries with the highest number of hospitalizations and deaths occurring in young children, mainly aged <5 years[2–4]. Although the currently available acellular pertussis (aP) vaccines have shown to be safe and efficacious, the recent increase in pertussis incidence in countries with well-established vaccination schedules and broad immunization coverage suggests that these vaccines are not providing long lasting control of the disease[5,6]. In light of this, there is an increasing interest in the development of new pertussis vaccines[7,8].

One of the most important virulence factors of *B. pertussis* and an obvious target for improving current aP vaccines is pertussis toxin (PTx). This exotoxin is a multimeric, 105 kDa protein, composed of five subunits arranged in a typical A–B structure[9]. In this holotoxin, the A-domain contains the enzymatically active S1 subunit while the B-oligomer, responsible for the binding to receptors on target cells, consists of two heterodimers, S2/S4-1 and S3/S4-2, coordinated by the S5 subunit[10]. PTx is an ADP-ribosylase that catalyzes the transfer of an ADP-ribose group from an $NAD^+$ substrate to a cysteine residue of the α-subunit of the trimeric GTP-binding protein (G protein)[11]. Ribosylation of these G proteins impairs the binding of G proteins to the G protein-coupled receptor, on the host cell membrane, promoting adenylate cyclase activity that causes an increase in the intracellular levels of cAMP. This PTx-driven accumulation of cAMP interferes with many cellular metabolic processes and induces a majority of the toxic effects associated with PTx such as leukocytosis, histamine sensitization, and alteration in insulin secretion by islet cells[12]. Animal studies have suggested that PTx contributes to the establishment of *B. pertussis* pathogenesis by hindering the migration of phagocytic cells to the site of infection, by suppressing early inflammation, and by inhibiting the microbicidal action of inflammatory cells[13]. In addition, production of PTx at the peak of *B. pertussis* infection correlates with exacerbated inflammation and pathology in the airways, confirming the key role played by this holotoxin in the pathogenicity of *B. pertussis*[14].

A chemically detoxified PTx, also known as pertussis toxoid (PTd), is included in all currently available aP vaccines. To eliminate the deleterious toxic effects, the toxin is inactivated by treatment with chemical agents, such as glutaraldehyde or formaldehyde. The chemical detoxification, while effective in disabling PTx toxicity, significantly impacts the overall structure of the toxin by impairing access to epitopes required for eliciting strong neutralizing antibodies against the toxin[15]. In an attempt to maintain similar conformational and immunological properties to that of PTx, while eliminating toxic effects, several genetically detoxified mutants of PTx have been generated for vaccine development[16–18]. From these, a variant with a double mutation in the S1 subunit substituting Arg9 by Lys and Glu129 by Gly (PT-9K/129G) has been shown to completely abolish the toxic effects and has displayed the lowest enzymatic activity of the many variants generated[17,18]. It has been hypothesized that the Arg9 substitution with Lys in PT-9K/129G interferes with the hydrogen bonding network of the surrounding residues, which disrupts the shape of catalytic pocket[15]. The Glu129 substitution by Gly has been implicated directly in diminishing the catalytic activity, either by impeding the electrostatic stabilization of a cationic transition state

intermediate or by preventing the deprotonation of the ADP-ribosyl acceptor substrate, which affects the cleavage of the *N*-glyosidic bond of the substrate $NAD^+$[19].

Genetically detoxified PTx (gdPT) is an attractive candidate for the development of improved aP vaccines based on the assumption that it will retain most of the structural properties of native PTx and therefore preserve native protective epitopes, compared to PTd. Significant progress has been made to better understand the immunological properties of gdPT[17,20,21]. Furthermore, two recent clinical trials that included gdPT in an aP vaccine revealed significant improvements in the antibody titers and duration of the responses[22,23]. Nevertheless, the impact of the double mutation on the overall structure, as well as the molecular basis for the lack of toxicity, is not completely understood.

In this manuscript, we present the complete crystal structure of the gdPT, in conjunction with hydrogen–deuterium exchange–mass spectrometry (HDX-MS), biolayer interferometry (BLI) of $NAD^+$ binding, and thermal denaturation to gain insight into the conformational differences between gdPT and PTx. We also use a previously unreported immune assay, based on mass cytometry, to compare the abilities of gdPT and PTd to restimulate a memory-immune recall response in a human model system. Taken together, these studies offer a unique approach to understanding the structure/function relationship that drives superior immunogenicity of gdPT.

## Results

**The structure of gdPT is nearly identical to that of PTx**. The crystal structure of gdPT was solved at 2.1 Å resolution (Protein Data Bank (PDB): 6RO0). The protein crystallized in a $P2_12_12_1$ space group with two gdPT molecules per asymmetric unit, consistent with the crystal formation reported for PTx[9,24]. We compared the structure of gdPT to the structure of PTx (PDB 1PRT)[9,24]. A schematic picture of the gdPT structure is shown in Fig. 1a and shows the typical A–B structure previously reported for PTx with the enzymatic A-domain (S1) resting on top of a disc-like base formed by B-oligomer composed of subunit S2, S3, 2 copies of S4, and S5 (Fig. 1). The overall structure and the arrangement of the subunits comprising gdPT was determined to be nearly identical to that of the PTx (Fig. 1b, c) with an overall root-mean-square deviation (RMSD) of 1.151 Å (when comparing with PDB 1PRT) and RMSD of 1.212 (when comparing with PDB 1BCP) for 1815 equivalent C-α atoms. Minor differences were, however, observed during structure refinement. Some of the loops in solvent-exposed regions show different conformations, in particular amino acids 45–50 in S5 subunit. This likely reflects the higher quality of diffraction data obtained, rather than true differences between the PTx and gdPT structures as the crystal packing in this region does not differ from the one in the models. In addition, the space group between PTx and gdPT was the same with nearly identical cell dimensions and crystal packing interfaces. Despite the minor differences observed between gdPT and the previously reported PTx structural data, the double mutation (R9K/E129G) in the catalytic site of subunit S1 (the A-domain) does not appear to affect the overall folding of the protein.

We then examined the structural difference of the catalytic site in the S1 subunit, where the two amino acid replacements in gdPT were made (Fig. 2). It has been hypothesized that the Arg9 substitution with Lys (R9K) in gdPT alters the hydrogen bonding network with surrounding residues, which then disrupts the shape of catalytic cavity[15]. In PTx, Arg9 forms hydrogen bonding interactions with Ser52 and Asp11 (Fig. 2c). The incorporation of Lys in position 9 maintains the hydrogen bonding with Ser52; however, it induced substantial changes in the hydrogen bonding network (Fig. 2d). Lys9 establishes

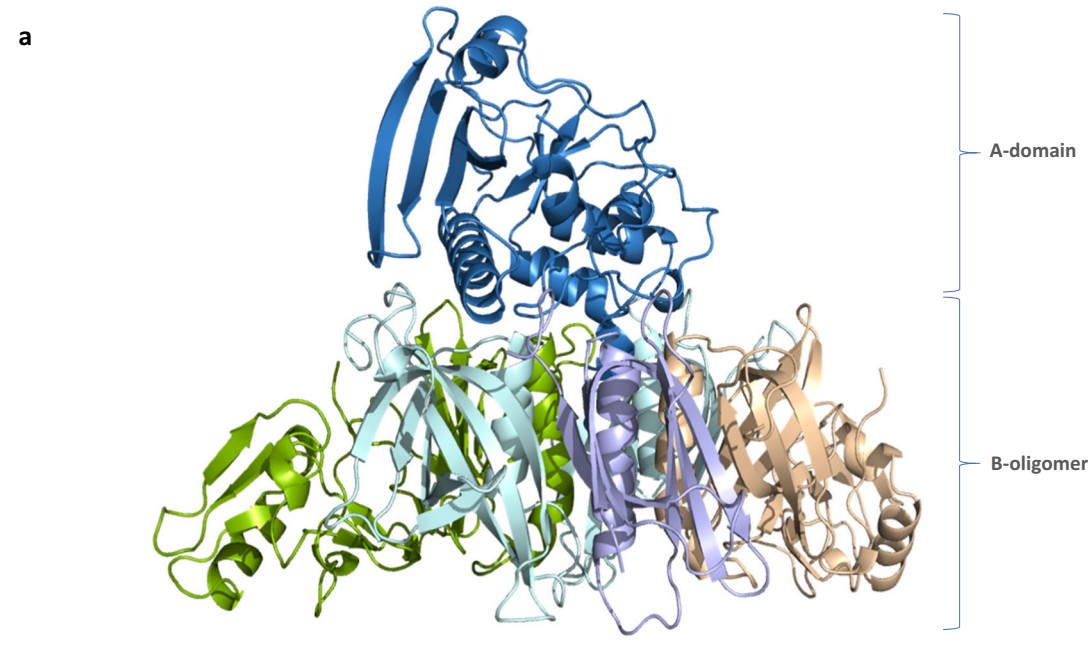

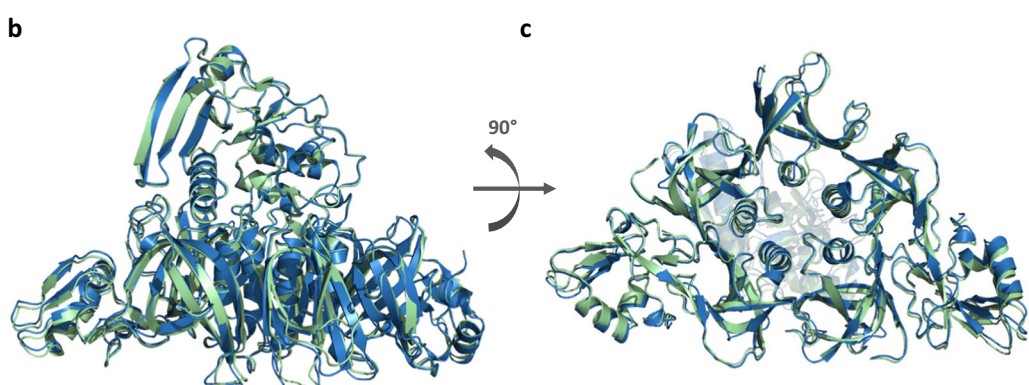

**Fig. 1 Comparison of PTx and gdPT X-ray crystal structures. a** X-ray structures of gdPT (PDB: 6RO0). Subunit colors: S1 blue, S2 beige, S3 green, S4 aqua, and S5 purple. **b** Comparison between gdPT (blue) and PTx (green) (PDB: 1PRT). **c** Bottom view comparison of gdPT and PTx.

hydrogen bonding with the carbonyl group of Tyr10, Gln205, and Met202 via water molecules, while the interaction with Asp11 is lost (Fig. 2c, d). When examining the effect of the E129G substitution, all interactions at position 129 (His35, as well as the hydrogen bonding with Gln127) were lost (Fig. 2e, f). The changes in the hydrogen bonding network due to either R9K or E129G mutations did not modify the shape of the catalytic pocket (Fig. 2). These results indicate that the double mutation does not alter the overall architecture of the active site, despite changes in the hydrogen bonding network surrounding Arg9 and Gly129 residues. However, it is unclear from the crystal structure whether the above changes would alter the conformational dynamics of the enzymatic site and consequently substrate binding.

**Double mutation in the A domain induces local and distal changes in gdPT conformational dynamics.** HDX-MS time course experiments were designed to assess any changes in dynamics and to provide key insights to better understand the effect of the mutations on the protein function and conformation. Digestion of non-deuterated control of PTx and gdPT yielded

average sequence coverage of 78% (Supplementary Fig. 1). Despite the high structural similarity between PTx and gdPT, HDX-MS experiments detected differences in dynamics behavior, measured by sequence-specific differences in deuterium uptake as shown in Fig. 3. A difference of 1.5 Da and 3σ (3× standard deviation of a given peptide over the five-point time course) was set as the statistically significant threshold for increase and decrease in deuterium uptake. When gdPT was compared to native PTx, the core of the A-domain (Subunit S1), specifically at the active site, showed changes in conformational dynamics, including an increase in deuterium uptake in the substrate binding cleft (Fig. 3a, red), as well as decreased deuterium uptake in the region where substrate access is controlled (Fig. 3a, blue), while sections of the B-oligomer also exhibited decreases in deuterium uptake (Fig. 3b, blue).

The A-domain of PTx is responsible for the catalytic activity of the protein complex. In gdPT, increases in deuterium uptake were detected in residues 1–17, 51–62, and 126–140 as shown in Fig. 4. The representative raw spectra for these residues are shown in Supplementary Fig. 2. These regions are located in catalytic cleft and make up the core of the A-domain. Regions with increased

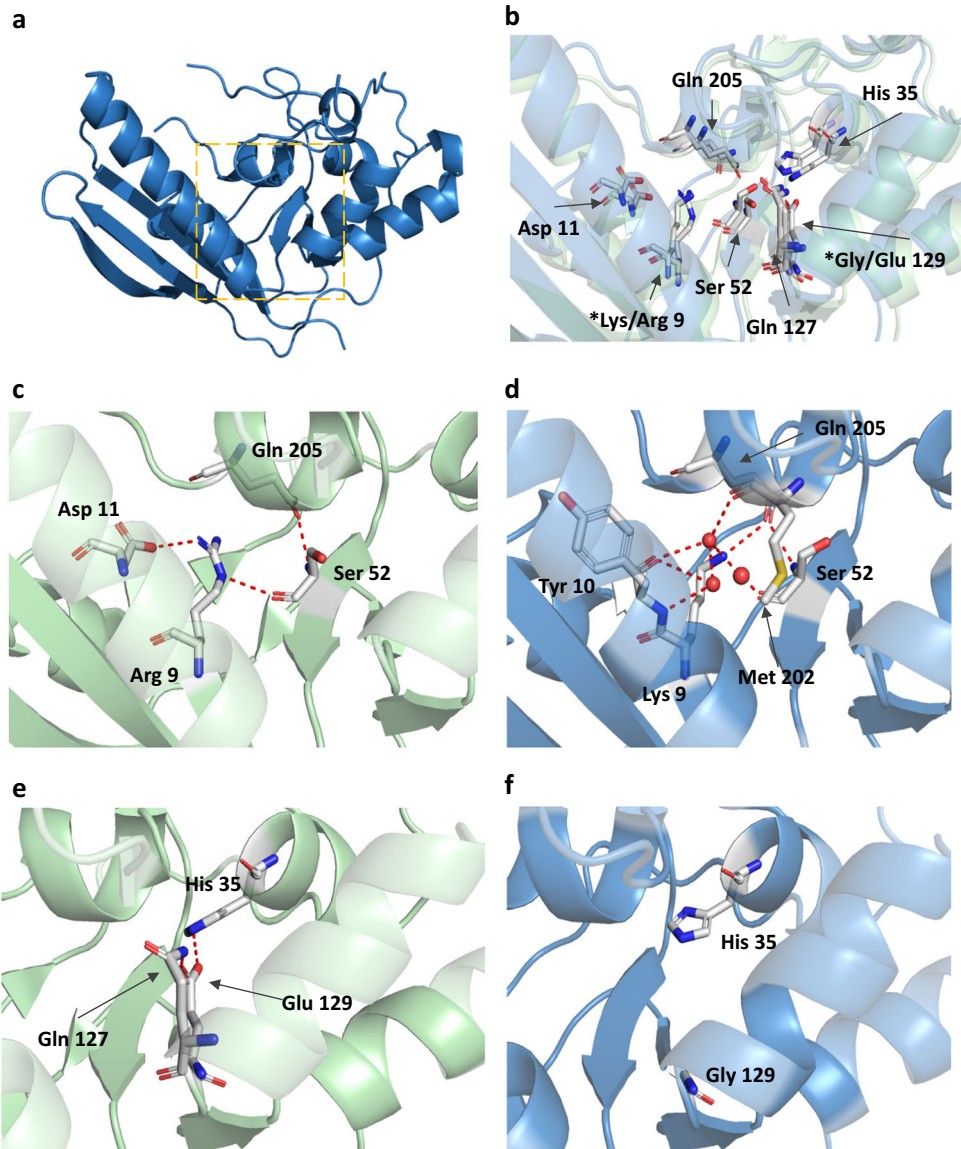

**Fig. 2 Detailed structural analysis of the A-domain catalytic site in subunit 1 of gdPT (PDB 6ROO) and PTx (PDB 1PRT). a** A-domain (Subunit S1) of gdPT with catalytic site indicated by orange box. **b** Superimposition of catalytic site of gdPT (blue) and PTx (green); asterisks (*) indicate mutation sites. **c** Hydrogen bonding network surrounding Arg9 in PTx. **d** R9K mutation in gdPT induces substantial changes in hydrogen bonding network. **e** Hydrogen bonding network surrounding Glu129 in PTx. **f** G129E mutation in gdPT disrupts hydrogen bonding network in this region.

deuterium uptake are more solvent accessible (Fig. 4a, red); this is a result of either greater structural flexibility in the protein sequence or changes in protein conformation. Since the crystal structure of gdPT did not show any differences in protein conformation, it can be inferred that the R9K and E129G mutations result in increased conformational flexibility in the catalytic core of the subunit.

The regions of increased conformational flexibility, sequences 1–17, 51–62, and 126–140, are defined by grouping of three β-strands located at the base of the catalytic cleft (Fig. 4a and the inset in Fig. 4b). Both the R9K and E129G mutations are contained within two of the three β-strand structure. HDX kinetic rates were obtained by fitting the uptake plots in Fig. 4b, using a single exponential parameter equation for gdPT and PTx peptides (calculations provided in Supplementary Table 1). Deuterium uptake rates are used to provide a measure of change in dynamic movements, detected at the peptide level, in specific regions where conformational differences are observed. The peptides

containing both mutation sites, residues 1–17 and 126–140, became 7.6× and 2.4× more flexible, respectively. The intrinsic rates of the 1–17 and 126–140 peptides were calculated for both gdPT vs. PTx sequences, and compared, to ensure that the observed dynamic differences for these peptides were solely from the secondary structure rather than the intrinsic rate difference, due to the mutation. For peptide residues 1–17 and 126–140, the ratios of the rates (gdPT/PTx) was 1.00 and 0.97, respectively. The tabulated rates and kinetic curve are provided in Supplementary Data 1. The peptide corresponding to residues 51–62, in the center of the three β-strands, became 4.4× more flexible. Arg9 is known to play a crucial role in stabilizing the catalytic cleft pocket through hydrogen bonding with Asp11 and Ser52 (Fig. 2d) to maintain the correct orientation for NAD$^+$ binding[9,15]. Glu129 is a critical residue in the ADP-ribosylation of NAD$^+$, where it forms a hydrogen bond network with His35, Ser52, and Gln127 (Fig. 2f)[9]. Increases in deuterium uptake in these regions imply that the gdPT core is dynamically different to that of PTx and

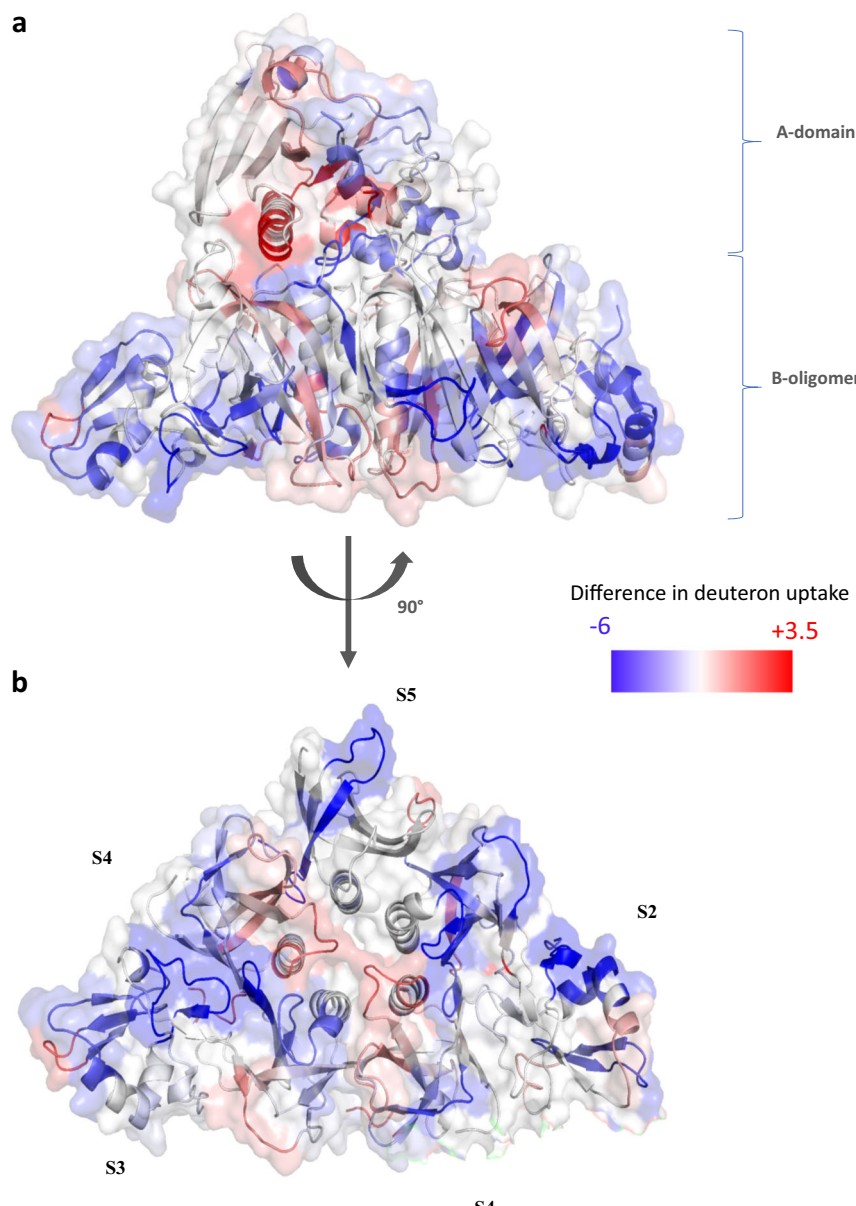

**Fig. 3 HDX difference plots mapped to the crystal structure of gdPT.** Differences in deuterium uptake were plotted onto the gdPT structure. A difference of 1.5 Da and 3σ (3× standard deviation of a given peptide/time point over the five time course) was set as the significant threshold for increase and decrease in deuterium uptake. Sequences with increased deuterium uptake are colored red, sequences with decreased deuterium uptake are colored blue, sequences in gray did not show any significant difference in deuterium uptake, and sequences in black were not detected in HDX-MS experiments. **a** Side view of gdPT and **b** bottom view of gdPT. For clarity, subunit 1 was omitted from the bottom view. Regions in gdPT that show increased deuterium uptake relative to PTx are plotted in red, while regions of decreased deuterium uptake are colored blue. Regions not covered by peptide mapping are indicated in black.

that although the static architecture of the domain is preserved the dynamic geometry necessary for NAD$^+$ binding is disrupted (Fig. 4b). A decrease in the rate of deuterium uptake for the 192–215 sequence was also observed, indicating a loss of flexibility in this region, which would further impact substrate access to the catalytic core of the S1 subunit (Fig. 4b and representative raw spectra are provided in Supplementary Fig. 2). This peptide corresponds to a helical loop that must be displaced during enzyme catalysis[9,15]. The rigidification of this mobile loop, along with increased flexibility in the catalytic cleft are predicted to impair NAD$^+$ binding within the catalytic site.

In addition, we observed changes in the dynamics of the B-oligomer, characterized by rigidification at subunit interfaces

(Fig. 3b and Supplementary Fig. 3). Specifically, the interfaces between S2/S4 (residues 113–141 of S2 and residues 82–97 of S4), S3/S4-2 (residues 15–31 of S3 and residues 82–97 of S4-2), and S4-2/S5 (residues 82–97 of S4-2 and residues 77–96 of S5) showed decrease in deuterium uptake. This indicates that the frequency of solvent exposure in these regions has decreased and indicates tighter packing among B-oligomer subunits.

Taken together, HDX-MS experiments highlighted substantial changes in the conformational dynamics of the catalytic site of the A-domain, in conjunction with rigidification of the subunit interfaces within the B-oligomer. These results also suggest that both NAD$^+$ interactions within the catalytic site, as well as conformational stability of the holotoxin B-oligomer, may be

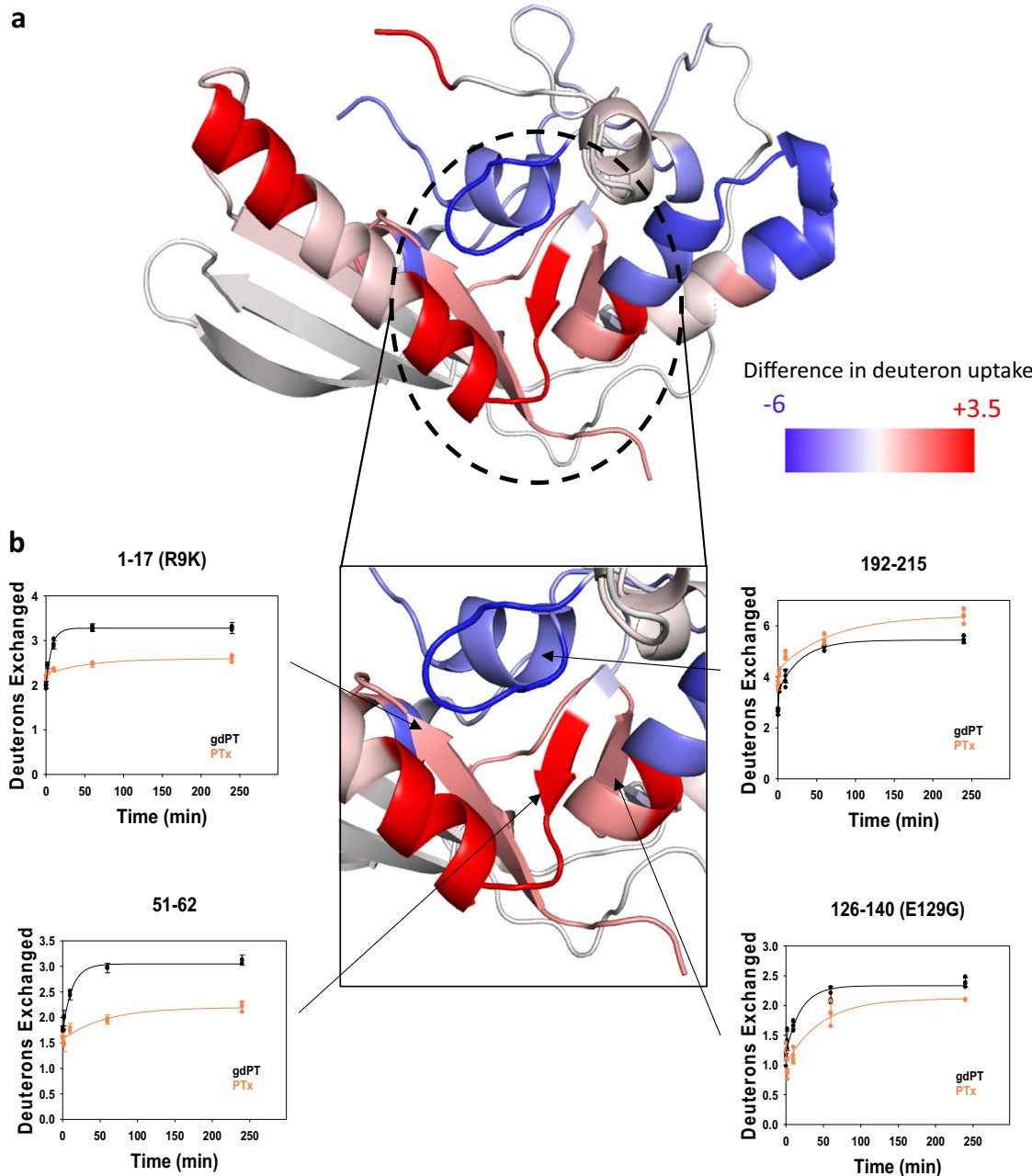

**Fig. 4 The catalytic site of gdPT subunit 1 is more flexible while the NAD⁺ shuffling loop is more rigid. a** HDX difference plots mapped onto the structure of subunit 1. A difference of 1.5 Da and 3σ (3× standard deviation of a given peptide/time point over the five time course) was set as the significant threshold for increase and decrease in deuterium uptake. Regions of subunit 1 that show increased deuterium uptake relative to PTx are plotted in red, while regions of decreased deuterium uptake are colored blue. Regions not covered by peptide mapping are indicated in black. **b** Inset detail of subunit 1 catalytic site. Kinetic plots of the affected sequences are also shown. Peptides containing the mutation sites: 1–17 and 126–140 showed increase in deuterium uptake compared to PTx. The peptide 51–62 that is known to stabilize the geometry of the active site also showed increase in deuterium uptake. The NAD⁺ shuffling loop (peptide 192–215) showed decrease in deuterium uptake.

impacted by the R9K/E129G mutations. This prompted us to investigate the NAD⁺ binding affinity and thermal stability of gdPT in comparison to PTx.

**Double mutation decreases substrate binding and improves thermal stability.** BLI was used to assess the impact of the gdPT double mutation on NAD⁺ binding in the S1 subunit (Fig. 5). The binding experiments conducted here are different from the conventional assays where $K_D$ is determined by the equilibrium reaction of the enzyme–substrate free in solution. Instead, biotinylated NAD⁺ was coupled to the biosensor and then incubated

with either gdPT or PTx in the first well, to measure association. Dissociation of the gdPT or PTx from NAD⁺ was measured by subsequent incubation of the complex in a buffer-only containing well (Fig. 5a). Three different concentrations of gdPT and PTx were tested: 3, 1, and 0.5 μM, as shown in Fig. 5b, c. At all the concentrations tested, the binding curves for gdPT produced lower amplitude signals compared to PTx, indicating lower binding affinity. $K_D$ values were determined from the association and dissociation curves and showed an increase in the $K_D$ for gdPT ($K_D = 5.5 \pm 2.9\,\mu M$) compared to PTx ($K_D = 0.9 \pm 0.07\,\mu M$). It is important to mention that the $K_D$ obtained for PTx by

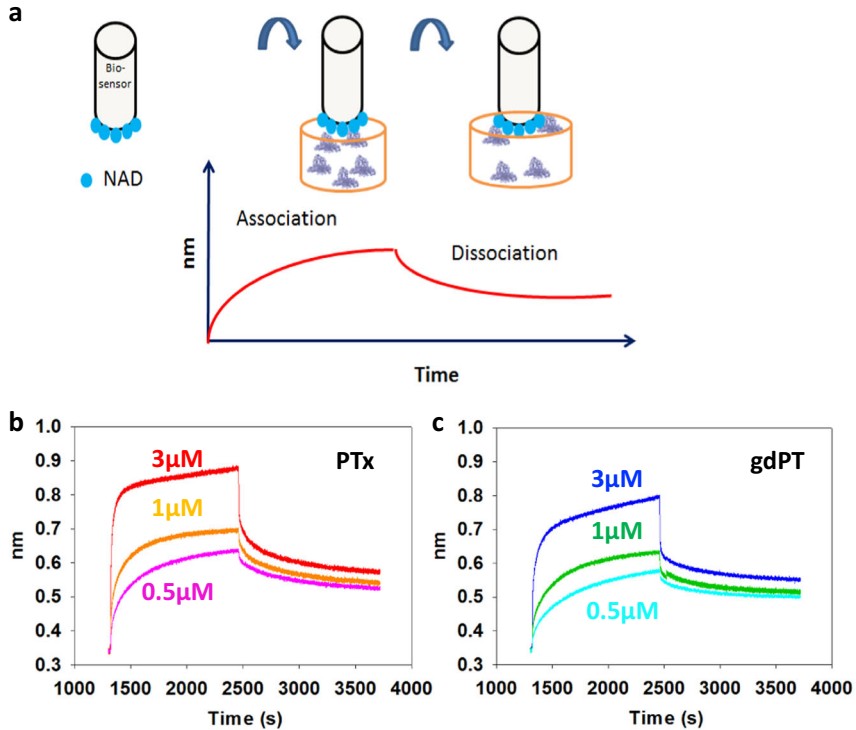

**Fig. 5 gdPT has lower binding affinity for NAD$^+$ compared to PTx. a** Schematic of the binding/dissociation assay using BLI. The biotinylated-NAD$^+$ was immobilized onto the streptavidin-coated bio-sensor, which was then allowed to bind to PTx- or gdPT-containing well, to establish an association (binding) curve. Subsequently, the complex was allowed to dissociate in a well containing buffer only, to obtain the dissociation curve. **b** Binding/dissociation curve of different concentrations (3, 1, 0.5 μM) of PTx to NAD is shown. **c** Binding/dissociation curve of different concentrations (3, 1, 0.5 μM) of gdPT to NAD$^+$ is shown.

BLI in this study was much lower than the previous value reported by Lobban et al.[25] (~25 μM), although that study used equilibrium of dialysis in buffer containing 1% urea. The lower $K_D$ (higher affinity) obtained for PTx may be due to the absence of urea in our experiments, which sustains a native conformation and better NAD$^+$ binding capacity. The kinetic parameters determined by binding studies are provided in Supplementary Table 2.

To investigate whether the double mutation in the A-domain would have an impact on the overall thermal stability of gdPT holotoxin, we compared the thermal unfolding of gdPT and PTx using SYPRO Orange extrinsic fluorescence spectroscopy; also known as differential scanning fluorimetry. The results showed a small, but statistically significant, increase in the melting temperature for gdPT when compared to that of PTx (ΔTm = 2.3 °C) indicating that the double mutation contributes to an increase in the overall stability of gdPT (Fig. 6). The effect of NAD$^+$ binding on the thermal stability was also investigated by SYPRO Orange extrinsic fluorescence but showed no substantial changes on the melting temperature for either gdPT or PTx even at high NAD:protein molar ratios (10:1) (Supplementary Fig. 4).

**gdPT induces a rigorous T cell memory response compared to PTd antigen in human model system**. The critical function of a vaccine antigen is to induce a protective immune response, and to boost the immune memory, to protect the recipient from subsequent pathogenic challenge. We have designed a previously unreported assay, based on mass cytometry, combined with a 30-marker panel that can monitor the memory immune response, after antigen stimulation, in a human model system. In this assay, whole blood from an aP vaccine-primed donor is used as a key

reagent to compare the impact of chemical detoxification (PTd) vs. genetic detoxification (gdPT) on their ability to stimulate the adaptive memory immune response, in vitro. Data visualization using viSNE clustering was applied to analyze and display high-dimensional data onto two-dimensional maps. Chemical detoxification of PTx significantly diminishes its immunogenicity by altering exposed epitopes[15]. Because the structure of gdPT is conserved relative to the native protein, conformational and linear epitopes are maintained, and the immune memory response is expected to be substantially broader relative to the chemically detoxified PTd.

We observed a substantial shift in cell phenotypes between PTd- and gdPT-stimulated cultures (Fig. 7a) but very little difference between PTd and the mock control (Fig. 7a and Supplementary Fig. 5). These results highlight gdPT's improved immunogenic characteristics, observed as a considerable shift from naive T cell populations to expanded memory T cells (Fig. 7a). The most noteworthy change observed between gdPT- and PTd/mock-stimulated samples was within memory T cell populations (CD45RO+). This substantial expansion of memory T cells in gdPT cultures present an activated phenotype, with increased expression of CD38 and CD25 in both the CD4 and CD8 compartments (Fig. 7 and Supplementary Table 3). These activated memory T cells were gated as CD38+ and CD25+, with a small subset further identified as HLA-DR+. In contrast, PTd cultures consisted mostly of resting memory T cells expressing low CD38, CD25, and HLA-DR (Fig. 7 and Supplementary Fig. 5). These results show that gdPT is inducing an expansion of activated memory T cells, compared to PTd stimulations. Activation markers expressed on memory T cells are better predictors of recall efficacy[26], compared to central and effector memory phenotypes.

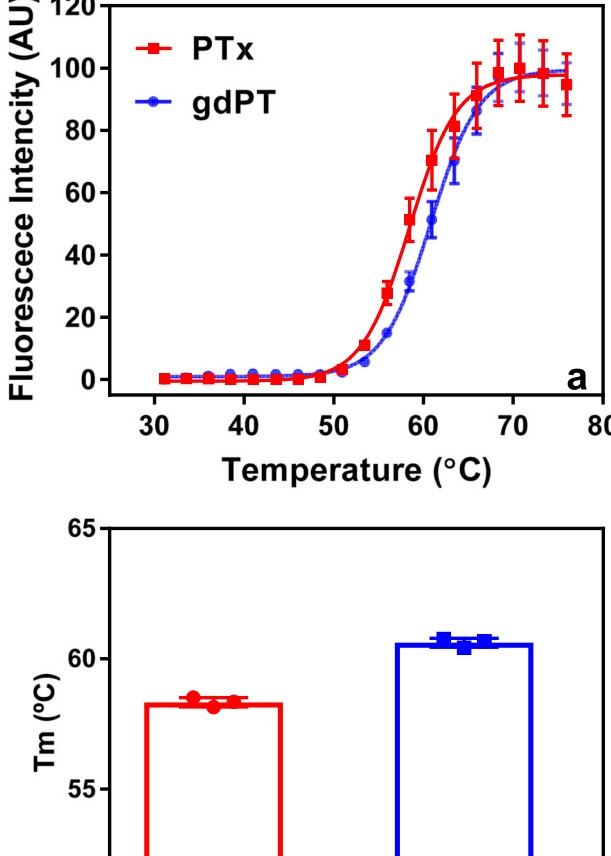

**Fig. 6 gdPT has increased thermostability compared to PTx. a** Thermal denaturation of gdPT and PTx was studied by collecting the SYPRO Orange extrinsic fluorescence as a function of temperature. **b** A significant increase in the thermostability of gdPT in comparison with PTx was observed with a $\Delta Tm = 2.1\,°C$ (Student's $t$ test, $p < 0.0001$, $n = 3$ independent experiments).

We also detected natural killer (NK) cells expressing CD45RO in gdPT-stimulated cultures, implying a memory-like phenotype in the NK subset. Similar to T cells, these NK "memory" cells have increased expression of HLA-DR and CD38 (Fig. 7a and Supplementary Fig. 5). This population is not present in PTd, or mock, stimulations. NK cells are traditionally classified as innate immune cells but can also act as mediators of the adaptive immune response, and CD45O+ NK cells have been identified in humans[27]. In addition, NK cells were shown to mount recall responses to whole-cell inactivated pertussis vaccination[28].

## Discussion

The correct structural functional relationship is critical to vaccine-mediated protective immunity. A vaccine must be processed by and presented to the immune system in a way that elicits an optimal response in a healthy human subject. Since its discovery, the non-toxic mutant gdPT was believed to be structurally similar to the wild-type toxin, with all conformational epitopes of native PTx preserved. This assumption was based on the ability of conformational and neutralizing antibodies to bind gdPT and by its superior toxin-neutralizing activity observed postimmunization[29,30].

The crystal structure of gdPT presented here unequivocally shows that the overall structure and the arrangement of the subunits are nearly identical to that of the native toxin. From this, we can infer that all conformational epitopes of wild-type toxin are conserved in gdPT. The most important feature of gdPT is the very low toxicity, which was believed to be induced, at least in part, by a disruption of the hydrogen bonding network and alteration of the architecture of the enzymatic cavity[17,18]. Our work also supports the observation that the mechanism of detoxification by amino acid replacement in gdPT is mediated, as least in part, by a disruption of the H-bond network, rather than through alteration of the structural architecture of the enzymatic cavity (Fig. 2). These results are consistent with a homologous genetically detoxified ribosyl transferase (CRM197) in which the organization/architecture of the $NAD^+$ binding pocket was unaltered[31].

To gain insight into the effect of decreased toxicity induced by the gdPT double mutation, we investigated the gdPT conformational dynamics by HDX-MS. Protein dynamics play a critical role in numerous functions, including catalysis, ligand binding, and protein/protein interactions[29,32,33]. Even subtle changes in dynamics can have a major impact on ligand binding[34] or cause a significant reduction in enzymatic funtion[31]. In this context, our HDX-MS data indicated rigidification of a mobile loop implicated with $NAD^+$ binding in conjunction with increased flexibility within the enzyme's catalytic cleft. These changes diminished $NAD^+$ binding to gdPT while simultaneously hampering enzymatic activity. Follow-up binding studies using BLI allowed us to focus on substrate–gdPT interactions and showed a fivefold decrease in $NAD^+$-binding affinity to gdPT when compared to that of PTx, confirming that the altered conformational dynamics of the active site has a direct influence on the ability of gdPT to bind its substrate.

Alongside the changes observed in vicinity of the mutation in the A-domain in gdPT, our HDX-MS experiments also revealed dynamic differences between subunit interfaces in the B-oligomer indicative of tighter subunit interactions. HDX-MS experiments provided direct detection of the conformational changes that result in tighter binding between subunits in the B-oligomer. Distal conformational changes in the B-oligomer were originally proposed by Ibsen who observed increased rates of binding for monoclonal antibodies to the B-oligomer of gdPT when compared to PTx[35]. It is not completely understood how mutational effects can propagate beyond the catalytic site toward the B-oligomer to cause changes in the conformation dynamics. Nevertheless, similar dynamically driven allosteric effects, which could not be anticipated from crystal structures, are not uncommon to proteins and have become increasingly reported in the literature[30,36,37]. It is highly plausible that the mutation-induced rigidification of subunit interfaces in the B-oligomer is responsible for both the gdPT spontaneous crystallization and its increased thermal stability compared to that of PTx.

The mass cytometric experiments addressed a crucial gap in the characterization of gdPT and bridged the structure of the protein directly to its function as an antigen. It is the structure of the antigen during its presentation to the immune system that is the critical attribute in the initiation of the memory immune response. This in vitro assay showed substantially higher memory T cell expansion and highlights the improved immunogenicity of gdPT antigen compared to that of PTd. PTx is a known mitogen with multiple receptor-binding and cross-linking capabilities giving it a dual role as both an antigen and an adjuvant[15]. The increased expansion of activated memory T cells and the appearance of memory-like NK cell populations in gdPT-stimulated cultures suggests that the immunogenicity effect associated with native PTx is maintained, supporting the overall

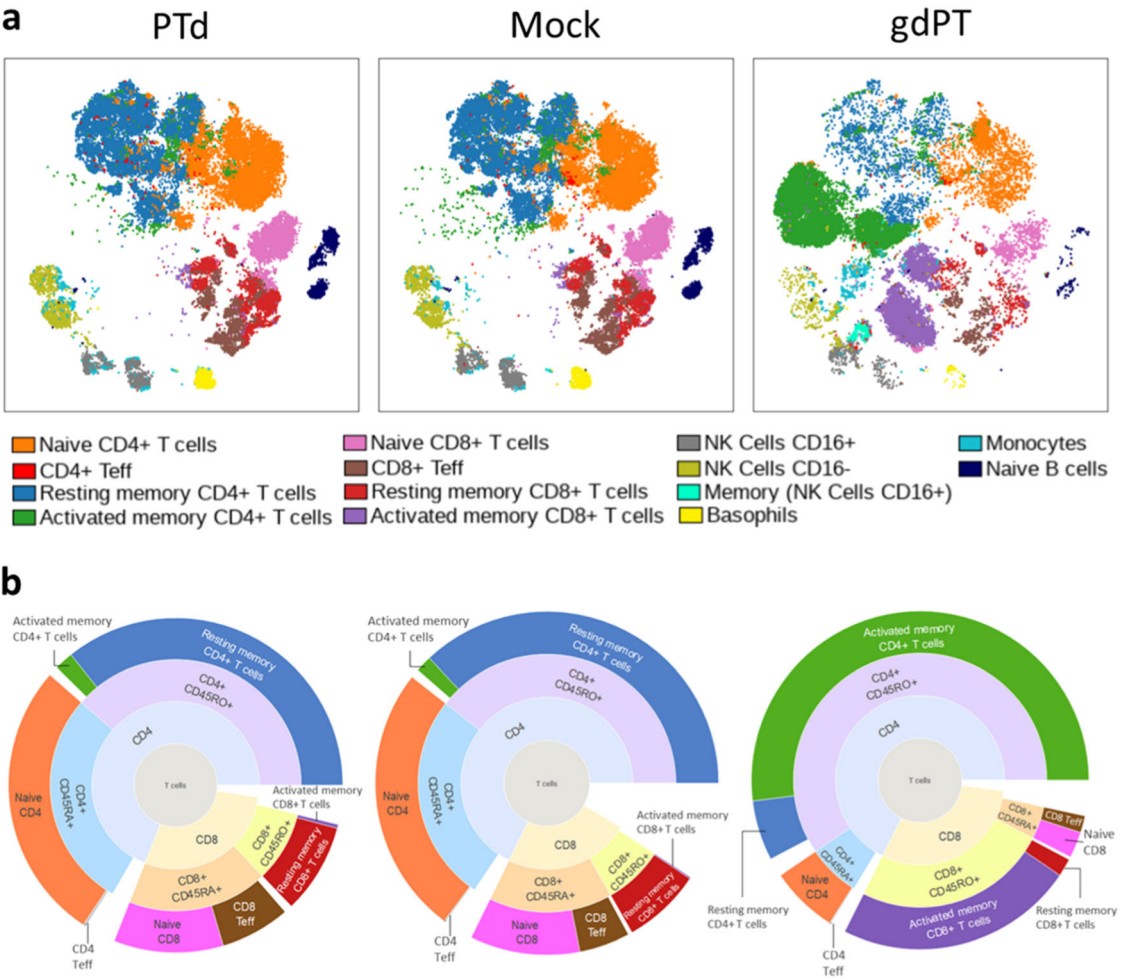

**Fig. 7 gdPT induces an activated memory response in an autologous human whole-blood (hWB) culture.** Mass cytometry was used to analyze whole-blood cells cultured for 7 days following stimulation with either PTd, gdPT, or a mock control (media alone). A tSNE-based dimension reduction analysis with viSNE was performed on CD45+CD66b− cells, showing the difference in cell population distribution following whole-blood stimulation. Every dot in the viSNE plot represents a single cell with an associated phenotype. **a** Cell populations defined by manual gating (Supplementary Fig. 6) projected onto the viSNE plots; each cell population is assigned a specific color. The CD123+CD11c− HLA-DR− pDC were identified and gated in viSNE plot based on the signal intensity of the cell surface markers CD123, CD11c, and HLA-DR. **b** A sun-burst plot representing the relative abundance (percentage of CD45+CD3+ events) of T cell populations following whole-blood stimulation. The concentric rings represent the gating hierarchies used and the sizes of the wedges are proportional to the abundance of the populations.

superiority of the gdPT antigen compared to PTd. The results from this assay are consistent with clinical observations of a relatively lower potency elicited with chemically detoxified PTx antigen and provides additional insight into how gdPT can induce a more robust immune response[15,22,23].

It should be noted that gdPT-containing vaccines are progressing or have progressed through clinical evaluation to licensure in combination with other pertussis antigens[15,22,23,38,39]. While these clinical studies have all demonstrated improved quality of the humoral response, relative to the chemically detoxified antigen, few have addressed the importance of the cellular response in detail. Our data use in vitro restimulation of adult human whole blood (hWB) to expose multiple early reactive phenotypes, including memory T cells, as important components of the response against gdPT. The data complement a wealth of information relating to superior antibody-mediated immunity and promotes continued clinical evaluation of the gdPT antigen.

Taken together, our results provide a more complete picture of the structure–function relationship that drives the noted improvements in gdPT immunogenicity. To our knowledge, the X-ray crystal structure shows for the first time that the structure

of gdPT is virtually identical to its native parent PTx. HDX-MS revealed that the functional activity differences between gdPT and PTx can be explained by mutation-induced changes in conformational dynamics of the active site and also provided further insight into distal changes within the B-oligomer that contributed to increased thermal stability and improved epitope presentation. Finally, mass cytometric experiments confirmed a direct link between gdPT structure and its enhanced immunogenicity, as measured by a substantial increase in memory T cell activation, and expansion, within the context of the adaptive recall immune response in a human model system.

## Methods

**Materials**. Purified PTx, PTd, and gdPT were produced by Sanofi Pasteur, Toronto (ON, Canada). Proteins were expressed in *B pertussis* and purified (~95%) by serial column chromatography following conventional purification protocols. gdPT was prepared at approximately 0.5 mg/mL in 10 mM potassium phosphate buffer, 150 mM NaCl, and 5% glycerol pH 7.0. PTx was prepared at approximately 0.5 mg/mL in 75 mM potassium phosphate buffer and 225 mM NaCl at pH 7.5.

**gdPT (R9K/E129G) crystallization, X-ray data collection, and structure refinement**. Crystals of gdPT spontaneously grew at 5 °C in plastic bottles containing

**Table 1 Data collection and refinement statistics (molecular replacement).**

|  | gdPT |
|---|---|
| Data collection |  |
| Space group | P2₁2₁2₁ |
| Cell dimensions |  |
| _a, b, c_ (Å) | 94.46, 160.79, 194.82 |
| _α, β, γ_ (°) | 90, 90, 90 |
| Resolution (Å) | 2.13 (2.2)ᵃ |
| $R_{sym}$ or $R_{merge}$ | 0.065 (1.95) |
| _I/σI_ | 11.5 (0.8) |
| Completeness (%) | 99.8 (99.5) |
| Redundancy | 4.8 (6.4) |
| Refinement |  |
| Resolution (Å) | 2.13 (2.2) |
| No. of reflections | 165,138 (11,816) |
| $R_{work}/R_{free}$ | 0.2600/0.2410 |
| No. of atoms |  |
| Protein | 14,506 |
| Ligand/ion | 54 |
| Water | 524 |
| _B_-factors |  |
| Protein | 64.9 |
| Ligand/ion | 93.5 |
| Water | 61.5 |
| R.m.s. deviations |  |
| Bond lengths (Å) | 0.007 |
| Bond angles (°) | 0.93 |

Values in parentheses are for the highest-resolution shell.
ᵃOne crystal was used to solve the structure.

0.5 mg/mL of protein in 10 mM KPi pH 7, 150 mM NaCl, and 5% glycerol. Also, 25% glycerol was included as cryoprotectant prior freezing. X-ray diffraction data were collected at the European Synchrotron Radiation Facility Beamline ID30A-1 with a Pilatus 3 2 M detector: wavelength of data collection = 0.966 Å, temperature of data collection = 100 °K, and beamline = ESRF Beamline MASSIF-1. Data were indexed/integrated using autoPROC[40], and molecular replacement was performed with Molrep[41] using PDB code: 1PRT as searching model. Model rebuilding was performed in Coot[42], and refinement was completed using autoBUSTER (version 2.11.7. Global Phasing Ltd., Cambridge, UK). Data collection and refinement statistics are listed in Table 1. Atomic coordinates have been deposited in the PDB under accession code 6RO0. Ramachandran favored/allowed/outliers (%) for gdPT were 97.9/2.0/0.1.

**Extrinsic fluorescence**. An Mx3005P™ Real-Time Polymerase Chain Reaction instrument (Stratagene, La Jolla, CA) was used to monitor the unfolding of PTx and gdPT. The buffer of gdPT was exchanged to the same buffer of PTx (75 mM potassium phosphate buffer and 225 mM NaCl at pH 7.5) using protein desalting spin column (Thermo Scientific, Rockford, IL). Protein unfolding was detected by the addition of the extrinsic dye SYPRO® Orange (Invitrogen, Carlsbad, CA) as described by Ausar et al.[43]. Briefly, 100 μL of each sample at 0.4 mg/mL was used to load 96-well polypropylene plates, and the plates were capped with optical cap strips (Stratagene, La Jolla, CA). The plates were centrifuged at 200 × g for 3 min prior to testing. The thermal profile used consisted of heating the plate from 25 to 90 °C at 1 °C/min. The fluorescence at 610 nm (excitation at 492 nm) was collected at every 1 °C increment. All experiments were performed in triplicate, and the data are presented as mean ± standard deviation. The melting temperature (Tm) for each sample was obtained by sigmoidal fit of the curves using GraphPad Prism 7.02.

**HDX mass spectrometry**. HDX-MS experiments were performed using the LEAP PAL automation technology coupled to Waters ultra-performance liquid chromatography (UPLC) and Synapt G2-S MS system as previously reported with minor modifications[44–46]. A detailed experimental workflow has been described elsewhere and only modifications were described here[46]. For non-deuterated and deuterated experiments, antigens were diluted in 10 mM potassium phosphate buffer (Sigma-Aldrich), pH 7.5, and deuteration buffer (10 mM potassium phosphate (Sigma-Aldrich), pD 7.5, respectively. Ten microliters of each sample was mixed with 30 μL of nondeuterated buffer, at 25 °C. Five time points were acquired for HDX experiments: 0.33, 2, 10, 60, and 240 min. Mixing in this ratio will attain maximum deuteration of 75%. Forty microliters of the reaction was quenched in 40 μL ice-cold quench buffer (100 mM potassium phosphate, 7.5 M GdnHCl, 0.5 M TCEP (Sigma-Aldrich), pH 2.5, 0 °C). The quenched samples were then diluted with 80 μL of 0.1% formic acid (Fisher Scientific). One hundred microliters of the

sample was then injected into the nanoACQUITY UPLC HDX module housing pepsin–protein XIII column (NovaBioAssays) for digestion. Subsequent digested peptides were desalted and separated for 3 min using 0.1% formic acid at a flow rate of 100 μL/min in the Waters BEH C18 VanGuard Pre-column and ACQUITY CSH C18 analytical column (Waters), respectively. The desalted peptides were then separated in the analytical column at a flow rate of 40 μL/min using a 7-min gradient starting from 1 to 60% organic phase (acetonitrile + 0.1% formic acid, Fisher Scientific). The eluted peptides were electrosprayed into and detected by the Waters Synapt G2-Si mass spectrometer (MA, USA) with the mass/charge (m/z) acquisition window of 300–1700. Intermittent spray of GluFib (785.8426 m/z, Sigma-Aldrich) was used as a lock mass solution to maintain mass calibration of <10 ppm. Blank injections (0.1% FA in LC-MS water) were incorporated in between HDX runs to prevent potential carry-over from previous runs.

**Peptide identification and HDX analysis**. Non-deuterated digested peptides were identified by mass accuracy and MS/MS fragmentation (Waters HDMSe function) using the ProteinLynx Global Server software (Waters Corp., MA, USA). Fragmentation (HDMSe) of non-deuterated peptides was activated in the transfer cell with the collision energy ramp from 15 to 65 V. The level of deuterium calculation on individual peptides was determined using the DynamX software (Waters Corp., MA, USA). HDX data were analyzed by calculating and summing the difference in deuterium uptake for identical peptides between the two states, PTx and gdPT, at all the HDX mixing time points. The data are all shown with respect to gdPT. Hence, positive or negative values indicate an increase or decrease in deuterium uptake in gdPT, whereas neutral value indicates no difference of the peptides. A difference of 1.5 Da and 3σ (3× standard deviation of a given peptide over the five time course) was set as the statistically significant threshold. A summary of HDX data corresponding to all peptides in the study is provided in Supplementary Data 1.

**HDX kinetic rate analysis**. The kinetic plots were made by plotting "deuterons exchanged" vs. "time." The rates were obtained by fitting exponential curve to the plot, $1 - e^{-xt}$, where x is the rate and t is the time. This equation represents single rate exponential curve, meaning one rate. For peptide residues 1–17, 51–62, and 126–140, a single exponential curve equation was used, whereas for peptide residues 192–215, a double exponential curve was used with equation $1 - e^{-xt} + 1 - e^{-yt}$ as single exponential equation did not provide a good fit. As a result, peptide residues 192–215 have a biphasic kinetic rate (x and y).

Intrinsic rates of peptides containing the two mutations, peptide residues 1–17 (R9K) and 126–140 (E129G), were compared to that of the PTx. To do so, the intrinsic rates of individual amino acid were obtained as established by Bai et al.[47]. The rate at individual time point was calculated by summation of the single exponential equation aforementioned for all the exchangeable hydrogens. For example, peptide residues 1–17 have 12 exchangeable hydrogens. As such, the rate will be summation of 12 single exponential equations at a specific time point. This step is re-iterated at different HDX time points. These data are plotted as function of time and the intrinsic rate was obtained by fitting the single exponential curve. The rate obtained for gdPT and PTx were then compared.

**HDX back-exchange analysis**. A back-exchange correction factor was not applied in the HDX differential analysis because comparisons were made on the same peptides from both protein states (gdPT vs. PTx). As such, the effect of back-exchange was considered minimal. Nevertheless, the back-exchange level on the HDX-MS system was characterized by analysis of 11 enolase peptides (Waters). The lyophilized peptides were reconstituted and equilibrated in deuteration buffer for 16 h. Subsequently, the sample was subjected to the same HDX-MS conditions as above and the resulting back-exchange is shown in Supplementary Table 4. On average, the back-exchange was calculated to be 29.9%[48–50].

**$K_D$ determination using BLI**. gdPT/PTx-NAD⁺-binding experiments were conducted using BLI on a FortéBio Octet RED384 system. The buffer used in the experiments was 50 mM Tris-HCl, 500 mM NaCl (Teknova) and 0.89 mg/mL bovine serum albumin (Thermo Fisher) at pH 7.4. The assays consisted of 5 steps: initial baseline of 60 s in the buffer followed by charging of the biosensor tip (FortéBio) with the biotinylated NAD⁺ (BPS Bioscience) for 1200 s. The probe was then washed and baselined for 60 s in the buffer with subsequent association step of 1140 s with either gdPT or PTx at concentrations of 3, 1, or 0.5 μM. The complex on the sensor was then moved to a well containing the buffer without any analytes for the dissociation step of 1250 s. $K_D$ was determined by calculating the ratio of $K_{off}$ and $K_a$, where $K_{off}$ is the dissociation rate and $K_a$ is the association rate. $K_a$ is determined by the equation: $(K_{obs} - K_{off})/[\text{protein}]$, where $K_{obs}$ is the rate constant at which the gdPT or PTx is binding to NAD⁺ (association step). Partial fitting equations were used to fit the curves where $Y = Y_0 + A(1 - e^{-K_{obs}t})$ and $X = X_0 + A(1 - e^{-K_{off}t})$ were used for $K_{obs}$ and $K_{off}$ determination, respectively.

**Mass cytometry assay**. _hWB assay_: Human blood used as a reagent in mass cytometry experiments was obtained from healthy donors (male or female) volunteers. Blood donors volunteered and were evaluated by registered nurses on-site at the Sanofi Pasteur Occupational Health Center (Toronto, Canada). Donors were screened

for infectious diseases and assessed independently by a physician while maintaining full anonymity. The volunteers completed and signed a statement giving informed consent permitting the use of their blood for research purposes. hWB stimulation was carried out as described by Hakimi et al.[51]. In brief, hWB from healthy adult donors was collected in BD Vacutainer® blood collection tubes containing heparin as an anticoagulant. Blood was diluted in serum-free Aim-V™ medium (Thermo-Fisher Scientific) and added to U-bottom 96-well plates. In all, 1 μg/mL of antigen (PTd, gdPT, or mock/media) was added to the hWB cultures. After a 7-day incubation, wells were pooled, and centrifuged at $300 \times g$. Cells were collected for mass cytometry antibody staining.

*Mass cytometry*: hWB cells were blocked in 100 U/mL sodium heparin salt to reduce nonspecific binding between metal-tagged antibodies and granulocytes[52]. Samples were then stained with a panel of 30 metal-tagged surface-marker antibodies (Maxpar® Direct™ Immune Profiling Assay™ System, Fluidigm). A list of the antibodies used can be found in Supplementary Table 5. Red blood cells were lysed in Cal-Lyse Lysing Solution (Thermo Fisher Scientific) according to the manufacturer's recommendations. Cells were washed in Maxpar® Cell Staining Buffer (Fluidigm). Cells were fixed in fresh 1.6% formaldehyde diluted in phosphate-buffered saline (Thermo Fisher Scientific), washed, and stained with Cell-ID™ Iridium-Intercalator diluted in Maxpar® Fix-Perm Buffer (Fluidigm) overnight. Cells were washed, counted, and resuspended in Maxpar® Cell Acquisition Solution containing 0.1× EQ™ Four-Element Calibration Beads (Fluidigm). Cells were filtered through a 35-μm mesh cap immediately prior to acquisition. Approximately 200,000 events per sample were acquired on a Helios™ CyTOF® System (Fluidigm) and normalized in the CyTOF software version 6.7.1014. The hWB cells used for mass cytometry were obtained from the same healthy donor.

*Single-cell analysis*: Data analysis was performed using Cytobank[53] by gating on intact cells based on the Cell-ID™ iridium DNA intercalators and live cells based on the Cell-ID™ rhodium DNA intercalator (Fluidigm). Manual gating on cell subset was performed according to Fluidigm recommended gating strategies (Supplementary Fig. 6). viSNE analysis was applied as a dimensionality reduction tool[54] reducing high parameter data down to two dimensions for easy visualization in Cytobank. viSNE analysis was carried out on the CD45+CD66b− cell population, using 300,000 desired total events (equal event sampling was selected), 3000 iterations, a perplexity of 100, and a theta of 0.5. Sun-burst plots were drawn using plotly package in the R software (Version 3.6.1; R Foundation for Statistical Computing, Austria).

**Statistics and reproducibility**. The results are expressed as mean ± standard deviation (SD). Statistical analysis when comparing two groups was performed using Student's $t$ test. When comparing data obtained from HDX experiments, a difference of 1.5 Da and $3\sigma$ (3× standard deviation of a given peptide/time point over the five time course) was set as the significant threshold for increase and decrease in deuterium uptake.

**Reporting summary**. Further information on research design is available in the Nature Research Reporting Summary linked to this article.

## Data availability

Atomic coordinates and related structure factors have been deposited in the Protein Data Bank with accession code 6RO0. Summary of all the peptides for both gdPT and PTx used for the HDX analysis is provided as Supplementary Data 1. The source data underlying Figs. 4, 5, and 6 are provided as Supplementary Data 2. Any other relevant data are available upon reasonable request.

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

## Acknowledgements
The authors thank Neil Blackburn, CMC Leader, R&D Product Portfolio Management at Sanofi Pasteur Ltd., Toronto for his helpful discussions and insightful review of this work. This project was supported in part by the Natural Sciences and Engineering Research Council (NSERC) of Canada Discovery through a Collaborative Research and Development (CRD) (485321-15) program. Funding for this study was also provided by Sanofi Pasteur, Canada both directly and as a partner in the NSERC-CRD grant above.

## Author contributions
D.A.J. and S.F.A. are responsible for overall design of this study. S.Z. and D.A.J. carried out HDX-MS and BLI experiments, data analysis, and interpretation. T.B., V.S., and A.R. analyzed and prepared for publication the X-ray structures of gdPT. S.F.A. and A.S. carried out extrinsic fluorescence experiments. M.C., J.D., and D.A.J. carried out mass cytometry experiment, data analysis, and interpretation. S.F.A., S.Z., J.D., R.H.B., and D.A.J. wrote the paper. D.J.W., R.H.B., and A.P. contributed to the paper preparation. All authors have given approval to the final version of the paper.

## Competing interests
S.F.A., A.S., R.H.B., A.P., and D.A.J. are employees of Sanofi Pasteur Ltd., Canada. T.B., V.S., and A.R. are employees of Sanofi France. S.Z. and M.C. were post doctoral fellows at York University at the time this study was executed. S.Z. is now with Sanofi Pasteur Ltd., Canada; M.C. is now at Fluidigm Canada. J.D. is currently a Post Doctoral Fellow at York University. S.L. is an employee of Fluidigm Canada. D.J.W. declares no competing interests.
