## [Peer Review File · Communications Biology]

Reviewers' comments:

Reviewer #1 (Remarks to the Author):

The manuscript by Ausar et al describes the structural and biochemical characterisation of a novel variant of pertussis toxin for use as a potential vaccine, with mass cytometry applied to study antigen stimulation. Overall, this work appears to be of high technical quality, and will be a useful contribution towards development of a novel pertussis vaccine. I have only minor corrections that can all be addressed with changes in text, with no additional experiments.

Major points

1. One of the main problems with the HDX data is that there is no comment in the results or figure legend for Fig. 3 of what a significant change actually is. There should be a clear statement in the results and figure legend as to what the threshold used for an increase or decrease in exchange (this is in methods, but inclusion in the text would ease interpretation). Also any statistical tests used to establish this should be mentioned (ie are they using 3 sigma from the average error of the entire experiment, or of a given peptide/timepoint?). Finally for both Figure 3 and 4 it might be useful to use different colors to establish the magnitude of difference. Right now a small and large change look the same on the structure .
2. The authors should include a HDX-MS data processing table as suggested by the recent community standards as shown in Masson et al Nature Methods 2019.
3. For the two peptides that show altered exchange that have mutations did the authors check the expected effect on the intrinsic rate of exchange as established by Bai et al from the Englander lab. This might be useful to establish the role of secondary structure vs intrinsic differences.

Minor points

1. The authors note minor conformational changes in surface exposed loops in this structure compared to previous (line 106). It might be useful to comment if these are in crystal packing regions of the structure as that could heavily influence the conformation. The authors also note the same space group, are other parameters the same between WT and mutant (ie unit cell dimensions and crystal packing interfaces?).
2. In figure 2 it would be much clearer if the atoms in the side chains were colored according to their identity (ie different colors for carbon, oxygen, and nitrogen for the sidechains). If using pymol, use color by element settings. It is always distracting to see a histidine ring all the same color. This will clarify the hydrogen bonds indicated by the authors, which is a major point of this figure.
3. Remove the A at the beginning of line 133.
4. The crystallographic data in Supp table 4 shows low I/Sig I, and a very high Rmerge in the outside shell. Could the authors include CC1/2 or CC* in the table to verify that data was truncated at the appropriate resolution?
5. The authors refer to SYPRO-orange extrinsic fluorescence spectroscopy (line 207), isn't this more frequently called Differential Scanning Fluorimetry?

Author report
John E Burke

Reviewer #2 (Remarks to the Author):

The manuscript by Ausar et al., is an elegant work on the characterization of structural and immunological properties of a genetically detoxified derivative of the pertussis toxin (gdPT R9K/E129G), which has been a major component of an acellular vaccine that was efficacious in efficacy trials and induced high levels of neutralizing antibodies in humans. These studies provide additional valuable insight into the structure-function relationship of gdP double mutant.

The 3D structure clearly shows that the overall structure and the arrangement of the subunits is nearly identical to that of the wild-type PTx. The introduction of the two mutations in the S1 subunit do not affect the folding, although the hydrogen bonding network undergoes to substantial changes, resulting in an increased conformational flexibility as measured by HDX-MS. The two mutations in the S1 subunit decrease the NadA + binding affinity. The B-oligomer seems to be influenced by the two mutations, resulting in a tighter packing of the other subunits, and a higher overall stability compared to the wild-type toxin.

The T-cell memory response was measured by mass cytometry after 7 days stimulation of whole blood from a donor vaccinated with the acellular pertussis vaccine. Although the data clearly show the higher expansion of memory T cells with an activated phenotype induced by stimulation with gdPT compared to PT chemically detoxified (PTd) , and that only gdPT is able to induce a memory-like phenotype in Natural Killer cells, further supporting the overall superiority of the gdPT compared to PTd , the conclusion are too stretched to support the assumption of stronger adaptive/memory response of gdPT and should be revisited.

Comments:

- The authors should discuss more the data from the previous clinical experience with gdPT, including trials in which the safety and immunogenicity of the two forms of PT have been compared e.g. Edwards KM, et al. Comparison of 13 acellular pertussis vaccines: overview and serologic response Pediatrics 1995; Greco D et al. A controlled trial of two acellular vaccines and one whole-cell vaccine against pertussis. N Engl J Med 1996;
- Line 72. Reference 18 (Pizza et al. 1989) should be added at the end of the sentence describing the discovery of the PT-9K/129G mutant
- Line 139 -142: "When gdPT was compared to native PTx, the core of the A-domain (Subunit S1), specifically at the active site, showed increase in deuterium uptake (Fig. 3a, red) while sections of the B-oligomer exhibited decreases in deuterium uptake (Fig. 3b, blue). Please rephrase the sentence that create ambiguity for the peptide 192 – 215 or change the color for this peptide
- Line 155-157: "HDX kinetic rates were obtained by fitting the uptake plots in Fig. 4b, using a single exponential parameter equation for gdPT and PTx peptides (calculations provided in Supplementary Table 1)." This is not clear how the values were obtained, please add extra details in material and method
- Line 176: S4-2 should be defined in line 48

Reviewer 1

Major points

Comment 1. One of the main problems with the HDX data is that there is no comment in the results or figure legend for Fig. 3 of what a significant change actually is. There should be a clear statement in the results and figure legend as to what the threshold used for an increase or decrease in exchange (this is in methods, but inclusion in the text would ease interpretation). Also any statistical tests used to establish this should be mentioned (ie are they using 3 sigma from the average error of the entire experiment, or of a given peptide/timepoint?). Finally for both Figure 3 and 4 it might be useful to use different colors to establish the magnitude of difference. Right now a small and large change look the same on the structure.

Response: The threshold used for increase/decrease in exchange was included in the figure legends (Figures 3 and 4) as well as in the results section: (line141-143) *“A difference of 1.5 Da and 3 σ (3x standard deviation of a given peptide over the five point time-course) was set as the significant threshold for increase and decrease in deuterium uptake”*. The statistical analysis of 3 sigma was also clarified by rephrasing. Figures 3 and 4 have been modified to reflect the magnitude of difference using blue-white-red colour gradient scheme.

Comment 2. The authors should include a HDX-MS data processing table as suggested by the recent community standards as shown in Masson et al Nature Methods 2019.

Response: As per request and to align with the HDX community standards, the PLGS and DynamX processing parameters are included in the supplementary excel file along with the HDX data summary, filename: HDXdataSummary_gdPT_v_PTx.xlsx.

Comment 3. For the two peptides that show altered exchange that have mutations did the authors check the expected effect on the intrinsic rate of exchange as established by Bai et al from the Englander lab. This might be useful to establish the role of secondary structure vs intrinsic differences.

Response: Thank you for the comment, this is a valid point and should be included in the manuscript. The intrinsic rates and plots for the peptides containing the mutation sites: Peptide 1-7 (R9K) and 126-140 (E129G) have been included in the excel spreadsheet for supplemental data (HDXdataSummary_gdPT_v_PTx.xlsx). The ratio of the wild type vs mutants was comparable, 1.00 and 0.97. Therefore, we concluded that the mutation did not alter the exchange rate of those peptides.

Minor points

Comment 4. The authors note minor conformational changes in surface exposed loops in this structure compared to previous (line 106). It might be useful to comment if these are in crystal packing regions of the structure as that could heavily influence the conformation. The authors also note the same space group, are other parameters the same between WT and mutant (i.e. unit cell dimensions and crystal packing interfaces?).

Response: Thank you for the comment. The crystal packing in the region displaying minor conformational differences does not differ from the one in the models. The space group between PTx and gdPT was the same with nearly identical cell dimensions and crystal packing interfaces. To address the reviewer's question, we added the following sentences to the revised manuscript (lines 110-112):

"This likely reflects the higher quality of diffraction data obtained, rather than true differences between the PTx and gdPT structures as the crystal packing in this region does not differ from the one in the models. In addition, the space group between PTx and gdPT was the same with nearly identical cell dimensions and crystal packing interfaces"

Comment 5. In figure 2 it would be much clearer if the atoms in the side chains were colored according to their identity (i.e. different colors for carbon, oxygen, and nitrogen for the sidechains). If using pymol, use color by element settings. It is always distracting to see a histidine ring all the same color. This will clarify the hydrogen bonds indicated by the authors, which is a major point of this figure.

Response: As per reviewer's suggestion, the amino acid side chains have been colored by atom in figure 2. Carbon, nitrogen, oxygen, and sulfur atoms are shown in white, blue, red, and orange respectively.

Comment 6. Remove the A at the beginning of line 133.

Response: *Corrected.*

Comment 7. The crystallographic data in Supp table 4 shows low I/Sig I, and a very high Rmerge in the outside shell. Could the authors include CC1/2 or CC* in the table to verify that data was truncated at the appropriate resolution?

Response: The CC1/2 was added to the supplementary table 4.

Comment 8. The authors refer to SYPRO-orange extrinsic fluorescence spectroscopy (line 207), isn't this more frequently called Differential Scanning Fluorimetry?

Response: The technique used in our study is extrinsic fluorescence as we have used an extrinsic dye (SYPRO-ORANGE) to study the conformational stability of the gdPT in comparison to PTx. This technique has been also referred to as thermal shift assay or differential scanning fluorimetry. Differential scanning fluorimetry is a general term used for thermal unfolding experiments using fluorescence (intrinsic or extrinsic). To address the reviewers question we have added a sentence to state that extrinsic fluorescence is also referred to as differential scanning fluorimetry (lines 218-219).

Reviewer 2

The T-cell memory response was measured by mass cytometry after 7 days stimulation of whole blood from a donor vaccinated with the acellular pertussis vaccine. Although the data clearly show the higher expansion of memory T cells with an activated phenotype induced by stimulation with gdPT compared to PT chemically detoxified (PTd), and that only gdPT is able to induce a memory-like phenotype in Natural Killer cells, further supporting the overall superiority of the gdPT compared to PTd, the conclusion are too stretched to support the assumption of stronger adaptive/memory response of gdPT and should be revisited.

Response: We deleted the sentence referring to a “stronger adaptive/memory response” and rewrote this section of the discussion with the following changes (lines 316-319): *...Supporting the overall superiority of the gdPT antigen compared to PTd. The results from this assay are consistent with clinical observations of diminished potency, elicited with chemically detoxified PTx antigen, and provides additional insight into how gdPT can induce a more robust immune response.*

Comments:

Comment 1 - The authors should discuss more the data from the previous clinical experience with gdPT, including trials in which the safety and immunogenicity of the two forms of PT have been compared e.g. Edwards KM, et al. Comparison of 13 acellular pertussis vaccines: overview and serologic response Pediatrics 1995; Greco D et al. A controlled trial of two acellular vaccines and one whole-cell vaccine against pertussis. N Engl J Med 1996

Response: To address the reviewer comment we have added a paragraph discussing our results with previously published clinical data comparing PTd vs gdPT (lines 320-327). The suggested references were also added to the manuscript.

“It should be noted that gdPT containing vaccines are progressing, or have progressed through clinical evaluation to licensure in combination with other pertussis antigens. While these clinical studies have all demonstrated improved quality of the humoral response, relative to the chemically detoxified antigen, few have addressed the importance of the cellular response in detail. Our data uses in vitro restimulation of adult human whole blood to expose multiple early reactive phenotypes, including memory T-cells, as important components of the response against gdPT. The data complements a wealth of information relating to superior antibody-mediated immunity and promotes continued clinical evaluation of the gdPT antigen”

Comment 2 - Line 72. Reference 18 (Pizza et al. 1989) should be added at the end of the sentence describing the discovery of the PT-9K/129G mutant.

Response: Reference 18 (Pizza et al. 1989) was added at the end of sentence describing the discovery of the PT-9K/129G mutant

Comment 3 - Line 139 -142: “When gdPT was compared to native PTx, the core of the A-domain (Subunit S1), specifically at the active site, showed increase in deuterium uptake (Fig. 3a, red) while

sections of the B-oligomer exhibited decreases in deuterium uptake (Fig. 3b, blue). Please rephrase the sentence that create ambiguity for the peptide 192 – 215 or change the color for this peptide.

Response: Reworded for clarity (lines 143-148): *When gdPT was compared to native PTx, the core of the A-domain (Subunit S1), specifically at the active site, showed changes in conformational dynamics including: an increase in deuterium uptake in the substrate binding cleft (Fig. 3a, red); as well as decreased deuterium uptake in the region where substrate access is controlled (Fig. 3a, blue). While sections of the B-oligomer also exhibited decreases in deuterium uptake (Fig. 3b, blue).*

Comment 4 - Line 155-157: "HDX kinetic rates were obtained by fitting the uptake plots in Fig. 4b, using a single exponential parameter equation for gdPT and PTx peptides (calculations provided in Supplementary Table 1)." This is not clear how the values were obtained, please add extra details in material and method.

Response: A Detailed explanation was added to the methods section under "HDX kinetic rate analysis" (lines 393-399). *The kinetic plots were made by plotting "deuterons exchanged" versus "time". The rates were obtained by fitting exponential curve to the plot, $y = 1 - e^{-x^*t}$, where y is amplitude of exchange, x is the rate and t is the time. This equation represents a single rate exponential curve, meaning one rate. For peptide residues 1-17, 51-62, and 126-140, a single exponential curve equation was used whereas for peptide residues 192-215, a double exponential curve was used with equation, $y = 1 - e^{-x^*t} + 1 - e^{-y^*t}$ as the single exponential equation did not provide a good fit. As a result, peptide residues 192-215 have a biphasic kinetic rate (x and y). Furthermore, detailed explanation of intrinsic rates comparison of peptides containing the mutation sites were also included.*

Comment 5 - Line 176: S4-2 should be defined in line 48.

Response: Corrected, see line 48: *consists of two heterodimers, S2/S4-1 and S3/S4-2*

REVIEWERS' COMMENTS:

Reviewer #1 (Remarks to the Author):

The authors have addressed all of my concerns, I fully recommend this article for publication.

John E Burke

Reviewer #2 (Remarks to the Author):

The authors have answered the questions this reviewer posed in a satisfactory manner. The work is now suitable for publication.